# VFaith: Do Large Multimodal Models Really Reason on Seen Images Rather than Previous Memories?

## Abstract

Recent works demonstrated that long-chain reasoning paradigms can enhance capabilities of multimodal large language models (MLLMs) to solve complex problems. However, the precise reasons for the effectiveness of such paradigms remain unclear and difficult to probe. Specifically, it is challenging to analyze with quantitative results how much the model's extraction of visual cues and reasoning during the long-chain inference process contribute to its performance improvements. Therefore, evaluating the faithfulness of MLLMs' reasoning to visual information is crucial. To address this issue, we first present a cue-driven automatic and instruction-following image editing pipeline with GPT-Image-1. Furthermore, we introduce VFaith-Bench, the first benchmark to our knowledge to evaluate MLLMs' visual faithfulness when generating long reasoning process. Using the designed pipeline, we constructed comparative question-answer pairs by editing the visual cues in images that are crucial for solving the original reasoning problem, thereby changing the question's answer to another option. By testing similar questions with images that have different details, the average accuracy reflects the model's visual reasoning ability, while the difference in accuracy before and after editing the test set images effectively reveals the model's faithfulness of reasoning to visual cues. We developed a filtering mechanism based on multi-model detection to identify error reason and self-contradictory within images. This approach, combined with manual verification, effectively eliminates image quality degradation. We conducted in-depth testing and analysis of existing mainstream flagship models and prominent open-source model series/reasoning models on VFaith-Bench, further investigating the underlying factors of their reasoning capabilities. Our code and data will be open-sourced after review period.

## 1 Introduction

With the advancement of multimodal large language models (MLLMs) (Li et al., 2024; Team et al., 2025; Wu et al., 2024; 2025; Zhu et al., 2025; Chen et al., 2025a; Team et al., 2023; Hurst et al., 2024; Guo et al., 2025b), the concepts of reasoning and slow-thinking have gained significant attention. Following the emergence of GPT-o1 (Jaech et al., 2024), numerous studies have explored the complex reasoning and extended thinking chains of MLLMs from various angles, including data synthesis and training methodologies. Approaches like InternVL MPO (Wang et al., 2024b), Llava-cot (Xu et al., 2024), which utilize structured Chain of Thought (CoT)(Lu et al., 2022; Wei et al., 2022; Yao et al., 2023) outputs, have achieved test-time scaling through structurally formed data, significantly enhancing the capability limits of MLLMs in tackling complex problems. With the growing popularity of DeepSeek R1 (Guo et al., 2025a), works such as Visual-RFT (Liu et al., 2025), VLAA-Thinking (Chen et al., 2025b), and Kimi k1.5 (Team et al., 2025) have integrated reinforcement learning algorithms like GRPO and DAPO (Yu et al., 2025) into MLLM training, further explored the capability limits of MLLMs in fields such as mathematics and coding. The combination of formalized rewards and reinforcement learning has become a common approach to train reasoning MLLMs.

Despite the establishment of reasoning patterns through structured data formatting and reinforcement learning, the precise mechanisms by which visual input and reasoning interact to augment large

model capabilities are still poorly defined or understood. Research on hallucinations (Li et al., 2023; Guan et al., 2024; Bai et al., 2024) in many MLLMs highlights a gap between the visual input and the reasoning process outputs in certain scenarios, indicating that the reasoning process of MLLMs may not always strictly adhere to the provided visual information.

But the frustrating reality is that analysis with quantitative results and comprehensive evaluations for assessing the visual fidelity of mainstream multimodal large reasoning models to their input are currently lacking. Furthermore, research on inconsistencies between reasoning and visual information in multimodal reasoning has been limited to manual case collection, lacking an efficient, large-scale pipeline for synthesizing erroneous reasoning data by attacking the reasoning process from a visual input perspective. This inefficiency hinders the progress of related research.

In response to this challenge, we propose VFaith-Bench, a benchmark built upon a cue-driven automatic and controllable editing pipeline that generates carefully constructed isomorphic problem pairs. These pairs feature visually similar inputs where subtle yet critical alterations to visual cues lead to different correct outcomes for the same query, thereby strictly probing models' reliance on visual evidence in their reasoning. This design is motivated and validated by the observation that humans, having understood the reasoning process and answer for a complex multimodal problem, can reliably solve these variants with near-perfect accuracy after only core visual modifications. Conversely, a significant performance drop in models on such pairs implies that their success on the original problems may not necessarily stem from true visual observation coupled with robust reasoning, but potentially from brittle patterns. Utilizing this benchmark, we conducted extensive evaluations to assess the visual fidelity and adherence capabilities of mainstream multimodal large reasoning models.

We applied our systematically designed cue-driven image editing pipeline to M3CoT (Chen et al., 2024b) and MegaBench (Chen et al., 2024a), two of the most recent comprehensive multimodal datasets, extracting questions from different distributions to construct a dataset consisting of 755 entries across six subsets. Through secondary manual verification, we ensured that the newly generated images are visually coherent and result in inconsistent standard answers with the original questions. This allows us to assess whether the models truly observe image-related cues and reason according to these cues. Using our constructed VFaith-Bench, we evaluated a range of prominent MLLMs, encompassing leading closed-source SOTA interfaces and popular open-source models with their reasoning variants. We found that:

- **Performance Degradation**: All models exhibited a significant average performance drop on questions featuring modified visual cues, indicating a high propensity for hallucination despite potentially coherent reasoning.

- **Perception Discrepancy**: Models exhibited significant hallucination in visual cue perception; a dedicated subset testing this yielded consistently low accuracy.

- **Pattern Adherence**: Hallucinations occurred even with modified benchmark data, where models often selected original, incorrect options, suggesting data leakage or over-reliance on training patterns instead of current visual information.

In summary, our work makes the following contributions:

- We developed VFaith-Bench (Figure 1), a benchmark to evaluate MLLMs' visual reasoning ability with an emphasis on the visual faithfulness, and conducted extensive evaluations of mainstream models.

- We introduce a cue-driven automatic and controllable editing pipeline (Figure 2), which is the first to leverage instruction-following image editing models for generating multimodal benchmark data specifically designed to probe model reasoning chains and induce hallucinations, and applied in our evaluations.

- Our evaluations of mainstream closed-source models, open-source models, and their reasoning variants revealed deficiencies in visual cue perception and adherence abilities, as well as potential data leakage and memory issues in existing benchmarks. These findings provide guidance for training more reliable multimodal large reasoning models in the future.

We believe this method, enabling the systematic creation of challenging test cases, holds significant potential as a key means for both diagnosing and advancing model capabilities in the long term.

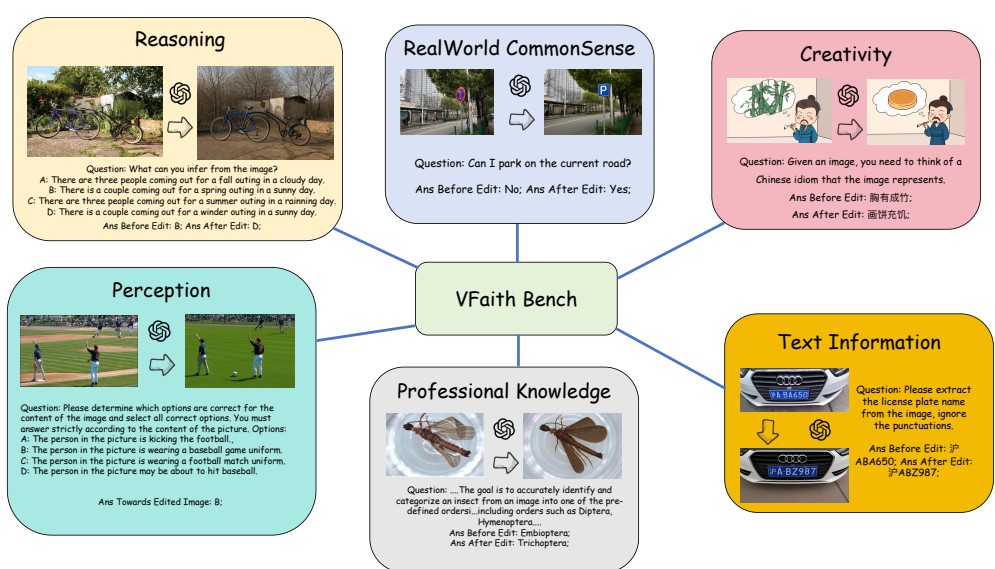

Figure 1: Overall view of VFaith-Bench. VFaith-Bench includes six subsets, 755 inputs. These are categorized into five subsets based on the content of the questions, along with a perception subset that directly queries the visual cues in the images. The image on the left side of each box is the original benchmark image, while the right is image edited by GPT-image-1.

## 2 RELATED WORK

**Multimodal Reasoning Benchmarks.** Advancements in Multimodal Large Models (MLLMs) have intensified focus on visual reasoning, spurring the development of cutting-edge multimodal reasoning benchmarks. These typically span diverse domains for comprehensive or domain-specific assessments. For instance, M3CoT comprehensively evaluates multimodal reasoning in science, common sense, and mathematics; MegaBench emphasizes real-world scenarios with 500 tasks; and benchmarks like MathVision (Wang et al., 2024a), WeMath (Qiao et al., 2024), and OlympiadBench (He et al., 2024) concentrate on detailed math and science reasoning. While these benchmarks have validated MLLM visual reasoning progress, they primarily explore existing capability boundaries. VFaith Bench differs by first evaluating visual reasoning across domains, then perturbing key visual cues via image editing. By analyzing changes in model responses pre- and post-perturbation, it aims to uncover the sources of MLLM visual reasoning improvements. Although some visual hallucination benchmarks assess visual understanding by altering visual information, their evaluations are often limited to simple comprehension judgments. They may not ascertain whether a model, within a reasoning context, accurately perceives visual cues rather than relying on data biases, which VFaith well addresses. We discuss more related multimodal reasoning methods and hallucination benchmarks in the appendix section A.2.

## 3 METHODOLOGY

### 3.1 CUE-DRIVEN AUTOMATIC AND CONTROLLABLE EDITING PIPELINE

Our process of synthesizing dual problems, illustrated in Figure 2, can be divided into the following three steps:

- **Step 1**: Extract visual cues towards origin question which relate to the groundtruth answer.

- **Step 2**: Generate rational modification suggestions to make original question have a new answer. Then invoke GPT-image-1 to complete image editing using the original image and the suggestions generated.

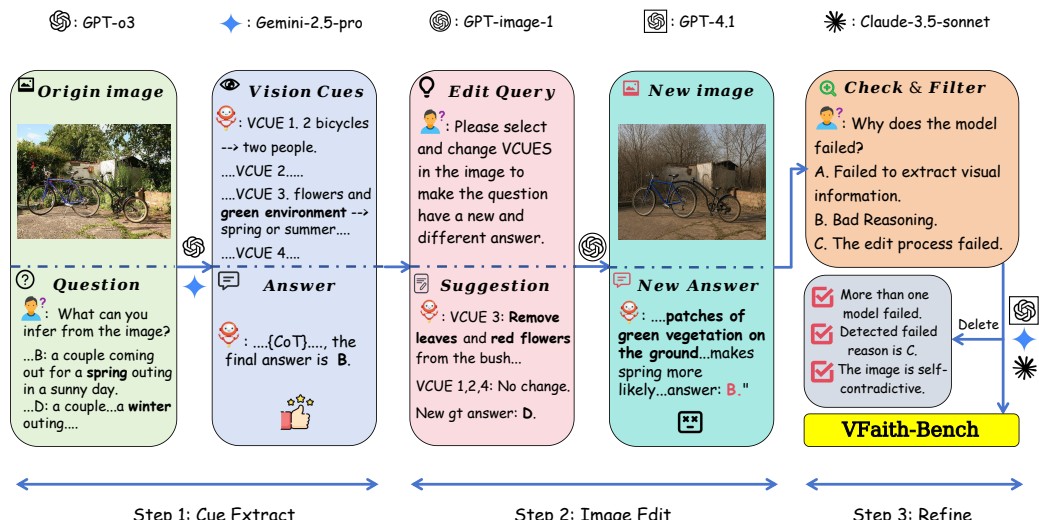

Figure 2: Our dual cue-driven image editing pipeline. First, SOTA reasoning models were employed to extract correct answers with long CoT and corresponding visual cues towards origin question. Then we used Gemini-2.5-pro to generate editing suggestions for useful visual cues, thereby guiding GPT-image-1 to perform edits on the original images, resulting in new correct answers. Finally, the newly generated image-question-answer triplets were used to evaluate models.

- **Step 3**: Refine the bench by deleting cases that fail to edit correctly to the suggestions and those have internal inconsistencies in edited images.

### 3.1.1 VISION REASONING WITH CUE

A critical aspect addressed early in this paper is how to get vision cues. We introduce a multimodal reasoning output style that differs from previous multimodal reasoning works (such as llava-cot (Xu et al., 2024), mulberry (Yao et al., 2024), vlm-r1 (Shen et al., 2025)). Previous works complete test-time scaling in different ways, like requiring models to output in a formalized, phased manner with observation, analysis or summarization (as seen in llava-cot), or like deepseek-r1 (Guo et al., 2025a), letting models concentrate the observation and reasoning process entirely within the tags before outputting a conclusion, without any guidance for the thinking process.

The former approach, relying on a formalized CoT, often restricts output diversity and yields an unclear connection between visual content and the generated summary. Conversely, the latter approach, lacking any guidance or restrictions, tends to produce CoT outputs with reduced readability, complicating the extraction of crucial visual cues.

To effectively address these issues, we propose a MLLM reasoning output paradigm similar to the vision clues style depicted in Figure 3. Integrating the two aforementioned methods, we minimize the formalization requirements for models during test-time scaling, allowing them to reason based on the question and image content itself in any format, free from structural constraints. Models simply need to mark any referenced visual cues using the format <vcues_*> </vcues_*>, with * representing a number. These cues are then used for visual cue extraction and edit. The <vcues_> markers for visual cues effectively help us quickly extract cues that are crucial for reasoning, greatly improving the success rate of generating editing suggestions in the subsequent steps.

To obtain an effective and reasonable reasoning process and vision cues for all the data in our benchmark, we tested several open-source and closed-source models in practice, including GPT-o3 and Gemini-2.5-pro. Ultimately, by synthesizing results from multiple models, we acquired candidate visual cues and reasoning outcomes for the images. An example is shown in the figure on the right. For specific prompts and more examples, please refer to the appendix section A.3.

base_image    GPT-image-1    jimeng.ai    kling.ai

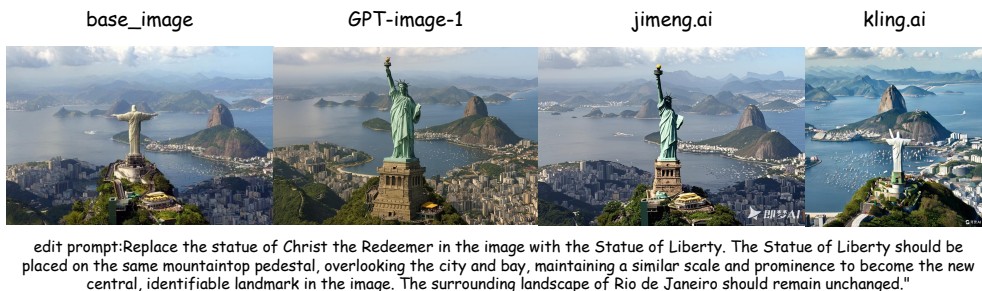

edit prompt:Replace the statue of Christ the Redeemer in the image with the Statue of Liberty. The Statue of Liberty should be placed on the same mountaintop pedestal, overlooking the city and bay, maintaining a similar scale and prominence to become the new central, identifiable landmark in the image. The surrounding landscape of Rio de Janeiro should remain unchanged."

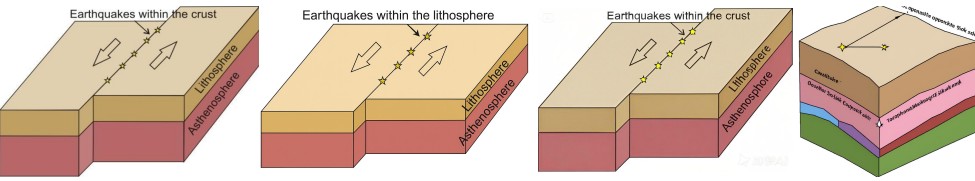

edit prompt:"Modify the text label 'Earthquakes within the crust' in the image to read 'Earthquakes within the lithosphere'. The stars, indicating earthquake locations, are already situated within the broader 'Lithosphere' layer. Changing this specific label will make 'lithosphere' the direct answer derived from the diagram's textual cue for earthquake location."

Figure 3: Performance comparison of different image editing models given editing instructions. GPT-image-1 provided editing results that better adhered to user instructions, with higher clarity, less text errors and best preservation of other features in origin image.

### 3.1.2 EDIT SUGGESTION WITH CUE

The high-performance image generation capabilities of GPT-image-1, have quickly set new benchmarks in the field of image editing. Through simple experimentation, we discovered that GPT-image-1 can already produce highly realistic images that are close to given instructions. We also compared GPT-image-1 with other SOTA image editing model in Figure 3 for certain examples to prove this.

Based on a thorough examination of an amount of carefully designed and selected cases, we believe that images visually modified by GPT-image-1 based on instructions can theoretically serve as approximate dual cases when used as new inputs for MLLMs. These cases can be used to explore various model capabilities, such as cue detection and reasoning coherence. However, the challenge remains in how to generate such cases in bulk. To address this issue, we have designed the following three principles:

- **Principle 1**: The selected vision_cues must be visual cues that directly influence the final answer, be clearly visible, and the corresponding modifications should clearly point to another answer among all the candidate options.
- **Principle 2**: The modification suggestions based on the selected vision_cues must be feasible and should not introduce any significant changes to the overall visual structure.
- **Principle 3**: The edited images, based on the selected vision_cues and reasonable modification suggestions, must not contain obvious common-sense errors.

Based on these principles, we completed the initial prompt development, as shown in Table 1. Detailed examples can be found in the appendix section A.4. Similarly, leveraging the latest closed-source MLLMs, we were able to generate reasonable editing suggestions in bulk for the reasoning results with vision cues produced in the previous sections.

We further designed methods to find low-quality cases caused by the limitations of the image editing models, like failing to follow edit instructions or edited images have self-contradictory. For further details, please refer to Section 4.4.

### 3.2 BENCHMARK OVERVIEW

### 3.2.1 BENCHMARK COMPOSITION

In previous section, we thoroughly presented the outcomes of various SOTA image editing models and interfaces on our selected data. However, reliable image editing still remains a challenge. For example, original answer may be linked to multiple visual cues, and partial editing of these cues results in ambiguous new image-text question pairs. This necessitates a manual review of the batch-generated cases. We developed an image quality checking pipeline to facilitate the rapid construction of the entire benchmark in Table 1.

Table 1: Dummy code of image quality checking

| **Image Quality Checking Pipeline** |
|---|
| maxtry = 5, try = 0, model = "Gemini-2.5-pro"; editor = "GPT-image-1",judger = "Claude-3.5"; input pic, q, vcue; while try < maxtry:    try ++;    edit_idea = model(vcue);    if judger(edit_idea, q, pic) == "pass":       pic_new = editor(pic, edit_idea);       if human(edit_idea, pic_new) == "pass":          return pic_new, edit_idea, model_ans; |

Based on the established suitability of M3CoT and MegaBench for challenging multimodal reasoning, we selected 664 entries to serve as the foundation for our dual problem pairs. The selection process intentionally excluded question types less amenable to visual modification, such as those requiring highly creative interpretation. The remaining 91 tasks in the benchmark are designed to assess the model's direct perceptual ability regarding the internal information of edited images. A detailed discussion of this will be presented in Section 3.2.2.

Using the methods introduced above, we generated corresponding dual images for all entries and manually verified that the same question towards edited image also has a correct results within the original options, and the original answer in non-edit dataset is no longer correct. We categorized these 664 entries into five categories based on content, as shown in Figure 1, with 235 Real World Common Sense(RWS), 132 Reasoning(REA), 169 Professional Knowledge(PFK), 57 Creativity(CRE) and 71 Text Information(TIF). The remaining perception task will be introduced in next section.

### 3.2.2 PERCEPTION TASK AND REPEAT RATIO METRIC

During evaluations, we observed a significant performance decline in current MLLMs on modified dual questions. Understanding the precise reasons for this decline is crucial, so we designed an additional **Perception Task** and a key metric **Repeat Ratio** to analyze these issues.

**Perception Task**: We selected 91 edited images and constructed questions focused on perceptual judgment by integrating original and modified visual cues. This type of question aims to directly assess the model's ability to get visual cues. We manually created multiple-choice options towards vision cues before and after edit to challenge the model to identify the correct descriptions of the modified visual cues.

**Repeat Ratio Metric**: Beyond errors caused by failing to interpret visual cues, we also seek to explore issues arising from the exact replication of the original question content and largely unchanged image content and structure. This could be due to the model's exposure to similar data during training, preventing it from breaking away from prior knowledge to produce accurate results. Alternatively, it could indicate a bias towards certain paradigms or difficulty in following specific instructions. The repeat ratio metric is calculated as follow, where $q$, $a_{ori}$, $a_{edit}$ means input question and the answer model generates before and after editing image, $gt_{ori}$ and $gt_{edit}$ means groundtruth answer to the queston before and after edit. $|\cdot|$ means the number of elements in a set. We hope this metric can indicate the proportion of hallucinations generated by the model's "memory".

$$Repeat\ Ratio = \frac{|\{(q, a_{ori}, a_{edit}) \mid a_{\text{edit}} = a_{ori} = gt_{ori}\}|}{|\{(q, a_{ori}, a_{edit}) \mid a_{\text{edit}} \neq gt_{edit}, a_{ori} = gt_{ori}\}|}$$

## 4 EXPERIMENTS

### 4.1 SETTINGS

**Evaluated Models**: For testing, models were categorized into closed-source APIs, including Gemini-2.5-pro, GPT-4o, Claude-3.7-Sonnet, and SEED1.5-VL, and open-source models divided into three

Table 2: Results of closed-source and large open-source Models. The relationship between abbreviations and category can be found in section 3.2.1. **Raw** and **Edit** means model's accuracy on dataset before and after editing, while $\Delta$, the key metric in our evaluation, is the change in the model's accuracy after editing images.

| Model | Metric | Type | | | | | | | Repeat | Perception |
|---|---|---|---|---|---|---|---|---|---|---|
| | | RWS | REA | PFK | CRE | TIF | Overall | **Refined** | | |
| Gemini-2.5-pro | Edit | 76.89 | 78.79 | 87.57 | 85.96 | 97.01 | **82.81** | **84.78**(±0.98) | 86.36 | 41.76 |
| | Raw | 89.92 | 86.36 | 88.76 | 85.96 | 94.12 | **89.01** | **89.20**(±1.16) | | |
| | $\Delta$ | -13.03 | -7.57 | -1.19 | +0.00 | +2.89 | -6.20 | -4.50 | | |
| GPT-4o | Edit | 71.01 | 67.42 | 81.07 | 73.68 | 95.59 | 75.60 | 76.64(±1.25) | 86.96 | 36.26 |
| | Raw | 81.09 | 74.24 | 84.02 | 77.19 | 94.12 | 81.48 | 81.28(±1.35) | | |
| | $\Delta$ | -10.08 | -6.82 | -2.95 | -3.51 | +1.47 | -5.88 | -4.64 | | |
| Seed1.5-VL | Edit | 74.37 | 71.97 | 84.02 | 71.93 | 97.06 | 78.46 | 80.96(±1.14) | 87.50 | **50.50** |
| | Raw | 88.24 | 84.09 | 86.98 | 61.40 | 94.12 | 85.39 | 85.60(±1.25) | | |
| | $\Delta$ | -13.87 | -12.12 | -2.96 | +10.53 | +2.94 | -6.93 | -4.64 | | |
| Claude-3.7 | Edit | 66.81 | 67.42 | 81.66 | 64.91 | 89.71 | 72.89 | 74.88(±1.31) | 72.92 | 37.36 |
| | Raw | 76.73 | 75.00 | 79.29 | 50.88 | 80.88 | 75.60 | 75.68(±1.39) | | |
| | $\Delta$ | -9.92 | -7.58 | +2.37 | +14.03 | +8.83 | **-2.71** | **-0.80** | | |
| Qwen2.5-72B | Edit | 67.65 | 65.91 | 74.56 | 49.12 | 91.18 | 69.88 | 70.72(±1.29) | 79.32 | 25.27 |
| | Raw | 78.15 | 78.79 | 72.19 | 71.93 | 85.29 | 76.96 | 77.44(±1.38) | | |
| | $\Delta$ | -10.50 | -12.88 | +2.37 | -22.81 | +5.89 | -7.08 | -6.72 | | |
| Qwen2.5-32B | Edit | 66.39 | 60.61 | 76.33 | 54.39 | 91.18 | 69.28 | 70.35(±1.39) | 85.71 | 28.57 |
| | Raw | 78.57 | 75.00 | 76.33 | 59.65 | 77.94 | 75.60 | 75.80(±1.47) | | |
| | $\Delta$ | -12.18 | -14.39 | +0.00 | -5.24 | +13.24 | -6.32 | -5.45 | | |
| InternVL3-78B | Edit | 60.92 | 55.30 | 58.58 | 45.61 | 85.29 | 60.39 | 61.12(±1.27) | 80.52 | 46.15 |
| | Raw | 83.19 | 67.42 | 79.29 | 70.18 | 86.76 | 78.31 | 78.08(±1.54) | | |
| | $\Delta$ | -22.27 | -12.12 | -20.71 | -24.57 | -1.47 | -17.92 | -16.96 | | |
| InternVL3-38B | Edit | 61.34 | 58.33 | 61.54 | 57.89 | 82.35 | 62.65 | 63.68(±1.41) | **73.13** | 40.66 |
| | Raw | 78.57 | 72.73 | 81.66 | 49.12 | 85.29 | 76.36 | 76.48(±1.54) | | |
| | $\Delta$ | -17.17 | -14.40 | -20.08 | +12.87 | -2.96 | -13.71 | -12.80 | | |
| Ovis2-34B | Edit | 68.07 | 63.64 | 66.27 | 62.07 | 89.71 | 68.42 | 69.33(±1.31) | 74.55 | 37.36 |
| | Raw | 81.51 | 78.03 | 66.27 | 64.91 | 91.18 | 76.51 | 76.96(±1.45) | | |
| | $\Delta$ | -13.44 | -14.39 | +0.00 | -2.84 | -1.47 | -8.09 | -7.63 | | |

Table 3: Results of all open-source models and smaller reasoning models

| Model on Overall | Raw | Edit | $\Delta$ | **Refined $\Delta$** | Repeat | Perception |
|---|---|---|---|---|---|---|
| InternVL3-78B | **78.31** | 60.39 | -17.92 | -16.96 | 80.52 | **46.15** |
| InternVL3-38B | 76.36 | 62.65 | -13.71 | -12.80 | **73.17** | 40.66 |
| Ovis2-34B | 76.51 | 68.42 | -8.09 | -7.63 | 74.55 | 37.36 |
| Qwen2.5-VL-72B | 76.96 | **69.88** | -7.08 | -6.72 | 79.32 | 25.27 |
| Qwen2.5-VL-32B | 75.60 | 69.28 | **-6.32** | **-5.45** | 85.71 | 28.57 |
| InternVL3-8B | **69.28** | 57.23 | -12.05 | -11.47 | 81.16 | 30.77 |
| InternVL2.5-8B-MPO | 64.91 | 57.08 | -7.83 | -7.04 | 80.60 | **34.07** |
| Qwen2.5-VL-7B | 67.62 | 60.24 | -7.38 | -6.08 | **68.85** | 16.48 |
| Ovis2-8B | 68.98 | 64.46 | -4.52 | -3.10 | 75.47 | 29.67 |
| Valley2-7B-DPO | 68.07 | **65.61** | **-2.46** | **-1.65** | 80.36 | 20.90 |
| VLAA-Thinker-Qwen2.5VL-7B | **71.99** | 65.66 | -6.33 | -6.42 | 84.48 | 31.87 |
| Llama-3.2V-11B-cot | 58.73 | 56.24 | -2.49 | -1.90 | **76.79** | 20.88 |
| Kimi-VL-A3B-Thinking | 67.32 | **66.57** | **-0.77** | **+0.96** | **76.79** | **37.36** |

categories: model series with varying scales (e.g., Qwen2.5-VL, InternVL3) to observe performance changes with size; lightweight models (e.g., InternVL2.5-8B-MPO, Valley2-DPO) added to the smallest versions to establish a baseline for slow-thinking models; and SOTA reasoning/slow-thinking models (e.g., Kimi-VL-A3B-Thinking) to evaluate their claimed robust reasoning performance.

**Evaluation Pipeline**: VFaith consists six categories, mainly divided into VQA and single-choice questions. We directly queried the models without providing any shots. All thinking models were set to thinking mode. When evaluating responses, we first specify \boxed{} as the output format, extract answers from it, and directly compare them with the standard answers. However, non-

multiple-choice questions are not suitable for direct matching. Therefore, for cases where direct answer extraction and comparison fail, we use Claude-3.5-Sonnet for further accuracy assessment.

## 4.2 MAIN RESULTS

### 4.2.1 CLOSED-SOURCE AND LARGE OPEN-SOURCE MODELS

Our evaluation results for closed-source and large open-source Models are presented in Table 2. Based on the analysis of these results, we draw the following conclusions:

**Closed-source models still lead multimodal reasoning**: Flagship closed-source models lead significantly in original results, robustness against reasoning cue attacks, and visual perception. Gemini-2.5-Pro performed the best, achieving an original accuracy of 89, with only a 6.2-point (4.5 after refining) drop after image editing. It also perform well in perception metrics. Additionally, Doubao's latest model surpasses both GPT-4o and Claude-3.7-sonnet.

**Closed-source models are more likely to be trapped in fixed thinking mode**: Closed-source models are more likely to rely on memorized knowledge when responding, as evidenced by a statistically higher repeat ratio. This phenomenon may be caused by larger parameters and more textual data during training, which results in lower attention to visual information.

### 4.2.2 ALL OPEN-SOURCE MODELS AND SMALLER REASONING MODELS

Our evaluation results for all open-source models and smaller reasoning models are presented in Table 3. Based on the analysis of these results, we draw the following conclusions:

**Series Model Analysis**:

- **Scaling law is still well indicated by VFaith.** Across models of varying scales, we can clearly observe the scaling law in the **Raw** and **Edit** metrics. However, the differences between the 72B and 32B models are not pronounced, which may relate to the complexity of the problems. Qwen series models perform well across all sizes in terms of both original results and resistance to attacks, but their perception is noticeably lower compared to models of the same size.

- **Memorization of existing test cases may appear in some finetuned models.** Comparing the performance of the Qwen2.5-VL-7B baseline and models tuned from Qwen2.5-VL-7B shows a significant increase in the repeat ratio, indicating that the current performance improvement may largely stem from memorizing these reasoning data and patterns, rather than internalizing perceptual enhancements into its reasoning.

- **Perception structure matters, but requires more reliable alignment to advance the reasoning.** InternVL with the biggest vision encoders of all open-source models shows clear perception dominance, with models detecting modified cues more effectively; however, the overall results tend to decline. This suggests a possible misalignment between modalities or perhaps insufficient reasoning capabilities within the Intern series.

**Small-sized Model Analysis**: Among open-source MLLMs, those ranked highly perform similarly across the benchmark. However, Ovis series stands out in tasks after edited, being the closest to reasoning models. Its repeat ratio is also relatively low. When considering reasoning models, Kimi-A3B excels in results, reasoning fidelity, and visual perception. It demonstrates the best performance against adversarial cue modifications.

## 4.3 ERROR REASON ANALYSIS

In this section, we designed an experiment to analyze reasons behind the model's failures. We selected several representative models and used Gemini-2.5-pro to perform automated analysis of the mistake causes in all error cases. To avoid bias, we use Claude-3.5-Sonnet to assess the Gemini-2.5-pro's reasons for errors. We defined the following three types of error reasons:

- **Reason 1.** The predicted answer misunderstood the visual information in the image.
- **Reason 2.** The predicted answer has an error in the reasoning process, which is not about visual information in the image.

Table 4: Distribution of 3 kinds of error reasons

| Model | Total Error | Reason 1 | Reason 2 | Reason 3 | Perception Score |
|---|---|---|---|---|---|
| Qwen2.5-VL-7B | 264 | 133 (50.4%) | 104 (39.4%) | 27 (10.2%) | 16.48 |
| Qwen2.5-VL-32B | 203 | 120 (**59.1%**) | 66 (32.5%) | 17 (8.4%) | 28.57 |
| GPT-4o | 162 | 85 (52.5%) | 62 (38.3%) | 15 (9.3%) | 36.26 |
| Claude-3.7 | 180 | 57 (31.7%) | 100 (**55.6%**) | 23 (12.8%) | 37.36 |
| Gemini-2.5-pro | 114 | 44 (38.6%) | 56 (49.1%) | 14 (12.3%) | **41.76** |

- **Reason 3.** The change from original image to edited image doesn't match the edit suggestion, meaning there is a problem in the image editing process. Or the new ground truth answer is wrong.

Table 4 shows the distribution of the three error reasons of different models based on our statistics. The results show several points worth noting:

- The proportion of errors answers attributable to issues arising from image editing is relatively small (~10%). This indicates that our image editing pipeline has a high level of reliability.

- Models with relatively lower basic capabilities have a higher error rate caused by insufficient visual cue perception. This is consistent with the **perception** metric we set. Models with lower perception scores have poorer visual information extraction capabilities, and consequently make more errors in answering questions due to mistakes in visual information extraction (Reason 1).

- Models with higher perception scores have strong visual capabilities, and their errors mainly stem from Reason 2, reasoning process. This means that these models' reasoning chains may lack robustness when facing varying visual inputs or not strictly follow visual information.

### 4.4 Image Editing Quality Control and Refine

During editing images, the decline in image quality is worth noting. Low-quality edited images can result in ambiguity, which affects the validity of the evaluation. Besides manual check in Table 1, we also selected all **Reason 3** examples from Table 4. If an example appears at least twice under Reason 3, it indicates a high probability that it's a low-quality case caused by errors during the image editing process. After this round of filtering, we identified 21 low-quality samples.

Another potential cause of data quality degradation is partial editing of images that leads to self-contradictory content within the images. To address this issue, we employed Gemini-2.5-pro, Claude-3.5-sonnet, and GPT-4.1 to jointly evaluate the degree of self-contradiction of all edited images. The scoring scale ranged from 0 to 10, where 0 indicates severe self-contradiction and 10 signifies complete consistency. 18 cases with an average score below 3 were removed.

The results after refine can be found in the **refined** columns of Tables 2 and 3. After refining, the average value of $\Delta$ decreased by about 1%, while the relative magnitudes of $\Delta$ among different models remained almost unchanged, demonstrating the robustness of our evaluation. We also estimated the standard deviation of each model's accuracy on the refined dataset of 716 cases by performing 1,000 bootstrap iterations with a size of 1,000, enhancing the statistical significance of the results. The prompt used for checking image quality is provided in appendix A.3.

### 5 Conclusion

To evaluate the faithfulness of MLLMs' reasoning to visual information, we introduced VFaith-Bench, a benchmark designed to probe visual reasoning via image editing. By extracting visual cues within VQA questions and use GPT-image-1 to edit images, we constructed dual problem pairs that are subtle different. VFaith contains six subsets including an additional perception task, utilizes metrics to assess hallucination, visual cue perception accuracy, and reasoning performance on these edited inputs. Our evaluation revealed that this approach effectively challenges current model CoTs. The findings show importance of making MLLMs accurately perceive visual information during reasoning, while also suggesting potential issues of data leakage and memorization in existing bench. Limitations like high editing time cost and insufficient fidelity in detail modifications on some tasks are discussed in Appendix A.5.

## 6 ETHICS STATEMENT

We hereby affirm that our dataset does not contain any content that may contravene ethical standards. In the manual verification phase, we ensured the ethical integrity of the dataset.

## 7 REPRODUCIBILITY STATEMENT

We affirm that the data synthesis, image editing, and quality control components employed in our study can be reproduced using the prompts provided in the appendix A.3 and the language models referenced in the main text. The foundational dataset of VFaith-Bench is derived from publicly available benchmarks mentioned in section 3.2.1, ensuring full reproducibility.

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

## A APPENDIX

### A.1 THE USE OF LARGE LANGUAGE MODELS

In this research, LLMs were used for polishing the written drafts of this article, generating clues, edit suggestion and edited images in our main method. We guarantee that LLMs were not used for idea generation, experiment design, reference generation or any other purposes.

### A.2 RELATED WORKS

**Multimodal reasoning methods.** With the advancement of text-based reasoning models, multimodal reasoning models have also seen parallel development. The implementation of multimodal large reasoning models primarily involves methods such as Rationale Construction (e.g., Video-of-thought (Fei et al., 2024) and IPVR (Chen et al., 2023)), extensive CoT data training (e.g., Visual-o1 (Ni et al., 2024) and Llava-CoT (Xu et al., 2024)), and reinforcement learning (e.g., R1-OneVision (Yang et al., 2025), G-Thinker (Zhan et al., 2025) and VLM-R1 (Shen et al., 2025)). These approaches have enabled multimodal large reasoning models to achieve step-by-step reasoning for complex problems and improvements in model performance metrics. However, there is still a lack of in-depth mechanistic research into the reasons behind these performance enhancements. The aim of VFaith-Bench is to determine whether the model's reasoning process genuinely reflects the input visual information or

if the responses are influenced by memorized patterns of specific image structures during training. This study is beneficial for assessing the faithfulness of MLLM reasoning to visual information, contributing to the development of more reliable and powerful multimodal large reasoning models.

**Hallucination Benchmarks.** In MLLMs, hallucinations typically refer to text responses generated by the model that are inconsistent with the image information. Previous reviews have categorized multimodal model hallucinations into three types: object category, object attribute, and object relation (Bai et al., 2024) , with our work primarily focusing on the latter two categories. Some evaluations of hallucinations in MLLMs already exist. Early benchmarks such as POPE (Li et al., 2023) and MME (Fu et al., 2024) primarily consist of simple Yes-or-No tasks, which are insufficient for testing the performance of more advanced MLLMs. While CIEM (Hu et al., 2023) automates hallucination evaluation using large language models, its automation is limited to question generation. Bingo (Cui et al., 2023) examines hallucinations caused by perturbations in input information but relies entirely on manual annotation for dual image generation rather than automation. MERLIM (Villa et al., 2023) employs edited images for more targeted evaluations, but its image edits are limited to object instance removal, lacking diversity. VFaith-Bench has established an automated dual data synthesis pipeline for cue extraction and targeted editing of the reasoning chain, achieving more diverse image edits. This allows for more precise attacks on the reasoning processes of inference models, significantly enhancing data diversity, attack specificity, and automation.

## A.3 PROMPTS

In this section, we present the prompts used in the cue-driven automatic and controllable editing pipeline, as well as those employed in the evaluation process. Figure 4 illustrates the prompt utilized during the generation of visual cues. Within this prompt, we instruct the model to format the reasoning process using <think></think> tags, and to annotate extracted visual cues with <vcues_i></vcues_i> tags. This facilitates subsequent modifications of visual cues during image editing. We provide the model with few-shot examples within the prompt to enhance its understanding of visual cue extraction.

Figure 5 displays the prompt used to instruct the model to provide suggestions for editing visual cues based on the input question, image, and origin visual cues. In this prompt, we require the model to propose modifications to the cues without excessively altering the image. Another restriction is that the original answer should become incorrect after editing and that there should be a unique new correct answer. At this stage, we also provide the model with few-shot prompts to assist in generating suggestions for editing the visual cues. During the generation process, the model first generates the new answer, then formulates suggestions for editing the image based on this answer. Finally, the output is formatted in JSON to facilitate the extraction of the new answer and editing suggestions for subsequent processes.

Figure 6 illustrates the prompt used during the evaluation part. This prompt requires the model to first generate a CoT based on the analysis of the question and the content of the input image, followed by providing an answer. Within the prompt, we constrain the model to respond strictly according to the content of the image, thereby minimizing potential hallucinations from the aspect of prompts. At this stage, we conduct zero-shot testing instead of providing few-shot examples to accurately assess the model's faithfulness to the visual cues. The final output of the model is formatted using \boxed{} to facilitate the extraction of answers for evaluation purposes.

Figures 7 and 8 present the prompts employed during the refinement process. The prompt illustrated in Figure 7 provides the original image, the edited image, editing suggestions, question, and the ground truth answers before and after editing, enabling the model to determine the specific reason for errors made by the answering model. In contrast, the prompt in Figure 8 directs the model to assign a score based solely on the degree of self-contradiction present in the edited image. Utilizing these two prompts, we successfully implemented an effective automated filtering procedure.

## A.4 CASE ANALYSIS

We have provided an anonymous GitHub repository containing the full benchmark, available at **https://anonymous.4open.science/r/VFaith-Anonymous-C891**. In this section, we present example data from VFaith-Bench, including examples from both a non-perception subset (Figure 9, Figure 10) and a perception subset (Figure 11, Figure 12). In our publicly available complete benchmark, a

sample from the non-perception subset includes a pair of original and edited images along with a single question. This question yields different answers when given the edited and unedited images as input. We evaluate various models' accuracy in answering this question before and after image editing on each non-perception subset. Conversely, in the perception subset, we include only the edited image and a corresponding question manually constructed by the authors. This question poses four options based on visual cues from the original image, and the model is directly queried. After editing, some options reflect changes in visual cues compared to the original image. We assess the models' accuracy in answering the manually constructed questions within the perception subset.

### A.5   LIMITATION

In this paper, we have developed an efficient dual data synthesis pipeline to assess the model's ability to adhere to visual information. However, our practical implementation has encountered several limitations. Most image editing models, such as GPT-image-1, are closed-source and impose a query per minute (QPM) restriction, which limits the speed of our data synthesis. The release of advanced open-source image editing models in the future could potentially help expand our dataset. Furthermore, the end-to-end process of generating and editing opinions through models faces security constraints that affect the quality and efficiency of data synthesis. For instance, to ensure security, models often generate standard placeholders like '123456' when editing strings such as phone numbers. Additionally, editing requests involving facial information may be rejected by image editing model APIs due to security restrictions. To ensure that our published dataset is responsible and free from harmful content, we have conducted a manual review of the benchmark data released.

### A.6   BROADER IMPACT

Our work provides a crucial tool for researchers and developers to analyze the faithfulness of visual information processing in these models. This has significant implications for the development of AI systems that require reliable integration of visual and textual data, such as in autonomous vehicles, medical diagnostics, and assistive technologies. Our cue-driven editing pipeline, leveraging advanced image editing techniques, not only aids in evaluating existing models but also sets a foundation for improving model training methodologies and synthesizing training data. Furthermore, by highlighting discrepancies in model reasoning and visual perception, our research encourages transparency and accountability in AI development, fostering trust and ethical standards in deploying AI solutions across various industries.

**Prompt Used During Generate Visual Cues Towards Origin Question.**

You are a Visual Reasoning Corrector and Annotator. Process input data with these rules:

1. **Format Extraction**:
- Always wrap reasoning in <think></think>
- Extract answer to <answer></answer> without prefixes and postfixes.

2. **Validate and correct visual cue**:
- Cross-check visual cues between:
a) Model's <think> content
b) Image context
- If visual cues mismatch:
- Correct the model's answer based on the following rules:
- Prioritize corrections in this order:
1) Image
- Update subsequent reasoning accordingly and ensure logical consistency

3. Annotate visual cues:
- Wrap each distinct visual cue in <vcues_*></vcues_*>
- Start numbering from 1 (* = 1,2...)
- Tag specific phrases, not entire paragraphs

4. **Output Constraints**:
- Maintain original reasoning contents and flow structure as possible as you can
- Ensure that all annotations (<think>, <answer>, <vcues_*>, <aha>) are properly formatted and inserted in the correct locations.
- Corrected reasoning in <think> tags
- Answer in <answer> tags
- Visual cues properly annotated
- Maintain original correct answer

Example 1: Input data:
<question> What feature does the flip phone shown in the image have?
A. It has a large touch screen display
B. It cannot be used in low light conditions
C. It is able to take pictures
D. It has facial recognition technology </question>

<correct answer> C </correct answer>

<Model_Infer> To determine the correct answer, let's analyze the image and the options provided:

1. **Option A: It has a large touch screen display** - The image shows a flip phone, which typically has a small screen. The screen visible in the image is small and not a touch screen. Therefore, this option is incorrect.

2. **Option B: It cannot be used in low light conditions** - The image does not provide any information about the phone's camera or its performance in low light. This option is speculative and cannot be confirmed from the image alone.

3. **Option C: It is able to take pictures** - The image shows a camera lens on the flip phone, which suggests that the phone is capable of taking pictures. This option is likely correct.

4. **Option D: It has facial recognition technology** - Facial recognition technology is a modern feature not typically found in flip phones. The image does not show any indication of this technology. Therefore, this option is incorrect.

Given the analysis, the most reasonable conclusion is that the flip phone shown in the image is able to take pictures. Answer: C </Model_Infer>

Output 1:
<think>To determine the correct answer, let's analyze the image and the options provided:

1. **Option A: It has a large touch screen display** - <vcues_1>The image shows a flip phone</vcues_1>, which typically has a small screen. <vcues_2>The screen visible in the image is small and not a touch screen</vcues_2>. Therefore, this option is incorrect.

2. **Option B: It cannot be used in low light conditions** - <vcues_3>The illuminated keys imply that it can be used in low light conditions. </vcues_3>. Therefore, this option is incorrect.

3. **Option C: It is able to take pictures** - <vcues_4>The image shows a camera lens on the flip phone</vcues_4>, which suggests that the phone is capable of taking pictures. This option is likely correct.

4. **Option D: It has facial recognition technology** - Facial recognition technology is a modern feature not typically found in flip phones. <vcues_5>The image does not show any indication of this technology</vcues_5>. Therefore, this option is incorrect.

Given the analysis, the most reasonable conclusion is that the flip phone shown in the image is able to take pictures.</think>
<answer>C</answer>

Figure 4: Few-shot prompt we used during generate visual cues towards origin question and answer.

**Prompt Used During Generate Visual Cues Towards Origin Question.**

Now I will give you a multiple-choice question and 2-4 options to choose from. The model should analyze the question based on the input image and select the most suitable option to answer the question. I will provide you with the thinking process of the model. Please observe which visual cues VCUE are used during the thinking process and try to modify the images to change the visual cues and thus alter the answer to the problem.

You need to pay extra attention to which objects are included in the visual cues mentioned during the reasoning process, and provide strategies such as deletion/modification to guide subsequent image modifications. You need to choose the most suitable option that differs from the original correct answer and output a modified image strategy. After applying this modification strategy, the correct answer to the problem should be changed to the most appropriate option you have chosen that is different from the original correct answer, and the original correct answer no longer holds true. Please note that your modifications to the image should not result in significant changes to the original image. If you are unable to make minor modifications, please choose other possible candidate options.

Here is an example you can refer to:
Original image:
<example image>
Original question:
——Start Question——
Question: What feature does the flip phone shown in the image have?
A: It has a large touch screen display, B: It cannot be used in low light conditions, C: It is able to take pictures, D: It has facial recognition technology ——End Question——
Original answer:
——Start Answer——
C
——End Answer——
Original reasoning process:
——Start reasoning process——
<think>To determine the correct answer, let's analyze the image and the options provided:
1. Option A: It has a large touch screen display
- <vcues_1>The image shows a flip phone</vcues_1>, which typically has a small screen. <vcues_2>The screen visible in the image is very small and not a touch screen</vcues_2>. Therefore, this option is incorrect.
2. Option B: It cannot be used in low light conditions
- <vcues_3>The illuminated keys visible in the image imply that the phone can be used in low light conditions</vcues_3>. Therefore, this option is incorrect.
3. Option C: It is able to take pictures
- <vcues_4>The image shows a camera lens on the flip phone</vcues_4>, which indicates that the phone is capable of taking pictures. This option is correct.
4. Option D: It has facial recognition technology
- Facial recognition technology is a modern feature not typically found in flip phones. <vcues_5>The image does not show any indication of facial recognition technology</vcues_5>. Therefore, this option is incorrect.
Given the analysis, the most reasonable conclusion is that the flip phone shown in the image is able to take pictures.
By the way, check that no deduction strays from a real image clue.
The conclusion fits with what's visible in the scene.</think>
<answer>C</answer>
——End reasoning process——

Output content:
``` json
{
"new_option": "B",
"suggestion": "Change the phone screen buttons in this picture to non luminous and remove the camera part from the image."
}
Below, please output image editing strategies and new candidate answers based on the input content I provided. The input content is as follows:
Original image:
<input image>
Original question and candidate options:
——Start Question——
<options>
——End Question——
Original answer:
<original answer>
——Start Answer——
Original reasoning process:
——Start reasoning process——
<cot>
——End reasoning process——
Please analyze the content of the image, the questions and candidates, and the original reasoning process I provided to you step by step. Finally, output your answer in the following output format:
``` json
{
"new_option": "The new candidate option you have chosen",
"suggestion": "The modification strategy you described in language for the original image resulted in the correct answer to the original question being changed to the new candidate selected above"
}
```

Figure 5: Few-shot prompt we used during generate edit suggestions towards visual cues.

---

**Prompt Used During Evaluate Models on VFaith-Bench.**

Below, I will provide you with a picture and a question. Please analyze the picture and question step by step and give out your answer. You should strictly follow the image and do not add any other information. {prompt_text}. Output the final answer in the format `\boxed`.

---

Figure 6: Zero-shot prompt we used during evaluate models on VFaith-Bench.

---

**Prompt to check the fail reason of models.**

You are an AI evaluation expert. Please analyze the following information:

1. Question:

<question>

2. Original image:

<ori_image>

3. Edit suggestion (description of the image modification):

<edit_suggestion>

4. Edited image:

<image>

5. Expected standard answer:

<new_gt_ans>

6. Model's predicted answer:

<predicted_answer>

Please analyze why the model's predicted answer is inconsistent with the expected standard answer.

Your attribution must be strictly limited to one of the following three categories:

1. The predicted_answer misunderstood the visual information in the image.

2. The predicted_answer has an error in the reasoning process, which is not about visual information in the image.

3. The change from "ori_image" to "edited_image" does not match the description in "edit_suggestion", meaning there is a problem in the image editing process. Or the new_gt_answer is wrong.

Please output your reasoning process first, then output the final attribution result enclosed exactly between the tags <answer> and </answer>.

Example format:

[Your detailed reasoning...]

<answer>

Only output attribution result ID(one of the three categories above)

</answer>

---

Figure 7: Prompt to check the fail reason of models.

**Prompt to check the self-contradictory in edited images.**

You are an AI evaluation expert. Please analyze the following information:

1. Question:

<question>

2. Original image:

<image>

3. Edit suggestion (description of the image modification):

<edit_suggestion>

4. Edited image:

<edited_image>

5. Expected standard answer:

<new_gt_ans>

6. Model's predicted answer:

<predicted_answer>

Please carefully analyze the content of the image before and after editing, determine whether there are any self-contradictions within the image after editing, and assign a severity score for these issues.

The score ranges from 0 to 10, where 10 means the edited image has no contradictory content, and 0 means the edited image has severe contradictions.

Please output your reasoning process first, then output the final score (0-10 integers) enclosed exactly between the tags <answer> and </answer>.

Example format:

[Your detailed reasoning...]

<answer>

Only output final conflict score (0-10), for example, <answer>2</answer>

</answer>

Figure 8: Prompt to check the self-contradictory in edited images.

---

**An example in non-perception subset.**

**Question**: Why does the front of the train appear dirty?
A: The train was caught in a rainstorm.
B: The train has not been washed in some time.
C: All options are possible.
D: The train was involved in a collision with another train. **Original answer**: B.

**Reasoning with visual cues**: <think>To determine why the front of the train appears dirty, let's analyze the image step by step:
1. Condition of the Train: <vcues_1>The front of the train has visible dirt and grime</vcues_1>, which suggests it has been in use for a while and has traveled through various conditions.
2. Environment: <vcues_2>The train is on a railway track with overhead electric lines</vcues_2>, indicating it's in a operational setting. The presence of dirt could be from the environment it has traveled through, such as dust, debris, or even road salt in some areas.
3. Maintenance: <vcues_3>The train appears to be in service</vcues_3>, and trains often accumulate dirt and grime over time, especially if they are not cleaned regularly.
4. Options Analysis:
- A. The train was caught in a rainstorm: While rain can cause dirt to accumulate, the image does not show signs of recent rain, such as wet surfaces or water streaks.
- B. The train has not been washed in some time: This is a plausible reason for the dirt on the train. Trains in service often accumulate dirt over time if not cleaned regularly.
- D. The train was involved in a collision with another train**: There is no visible damage or signs of a collision in the image.
Given these observations, the most likely reason for the dirt on the train is that it has not been washed in some time. Ensure all visual details are correctly recognized and that all reasoning steps are sound. Visual cues line up nicely with the reasoning steps.</think>
<answer>B</answer>

**Edit image suggestion**: Add vertical water streaks running down the dirty front surface of the train (<vcues_1>) and darken some patches of the gravel ballast near the tracks (<vcues_2>) to suggest dampness, implying the dirt is due to a recent rainstorm."

**New answer**: A.

Figure 9: An example in non-perception subset.

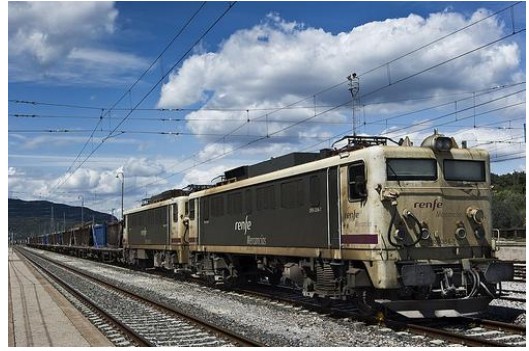 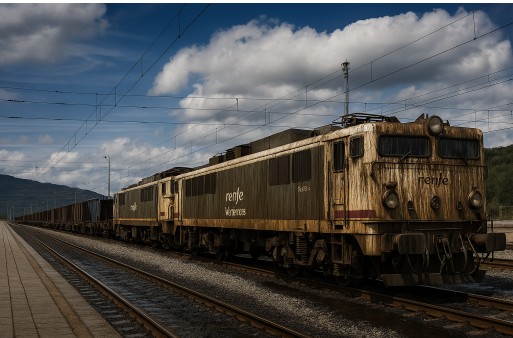

Figure 10: The corresponding example image in Figure 9. The left side is the original image, and the right side is the edited image.

---

**An example in perception subset.**

**Analysis with visual cues**: <think>To determine the setting of the scene, let's analyze the details in the image:
1. <vcues_1>The room has a bed with a cat lying on it, which suggests a personal living space</vcues_1>.
2. <vcues_2>There is a television on a stand with two lamps on either side, indicating a setup for watching TV</vcues_2>.
3. <vcues_3>The blinds on the windows are closed, and the room is lit by the lamps</vcues_3>, which suggests it might be nighttime or the room is meant to be cozy and dimly lit.
4. <vcues_4>The presence of personal items like a cat, a bed, and a TV setup indicates a residential setting, not a hotel room</vcues_4>.
5. <vcues_5>The room appears to be small and compact, which is more indicative of an apartment or a small house rather than a big house</vcues_5>.

Given these observations, the most reasonable inference is that the setting is a small residential room, likely a bedroom in an apartment or a small house.
Ensure all visual details are correctly recognized and that all deductions are sound.Visual cues line up nicely with the reasoning steps.</think>

**Edit image suggestion**: Modify the image to show the blinds on the window/door are open or partially open, revealing bright daylight outside. Also, edit the two lamps on the TV stand so they appear to be turned off (remove the glow)."

**Perception question**: Below, I will provide you with an image and four options. Please determine which options are correct for the content of the image and select all correct options. You must answer strictly according to the content of the picture. There may be any number of correct options, please put your answer in \boxed. If you have selected multiple options, directly output all correct option numbers in \boxed, such as AB, BCD, without adding spaces or any other content other than the letters ABCD.
Options:
A: The room has a bed with a cat lying on it, and a sofa on the right side.
B: The blinds on the windows are closed.
C: There is a television on a stand with two lamps opening on either side, indicating a setup for watching TV.
D: The room looks big and luxurious.

**Groundtruth answer**: A.

---

Figure 11: An example in perception subset. Note that the perception subset used in actual testing only includes the edited image and a question about the details of that image. In this example, options A, B, C, and D are asked for visual cues 1, 3, 2, and 5 respectively, where the visual cues corresponding to B and C have been edited.

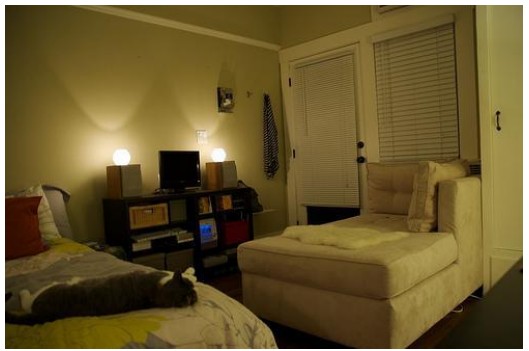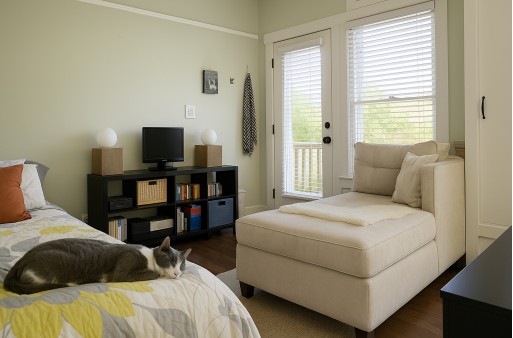

Figure 12: The corresponding example image in Figure 11. The left side is the original image, and the right side is the edited image.

