# OpenReview forum: "VFaith: Do Large Multimodal Models Really Reason on Seen Images Rather than Previous Memories?"
_ICLR.cc/2026/Conference — ICLR 2026 Conference Withdrawn Submission_

### Official Review · Reviewer_sYoe · 2025-10-27

**Soundness:** 2
**Presentation:** 2
**Contribution:** 2
**Rating:** 2
**Confidence:** 4

**Summary:**

This paper introduces VFaith-Bench, a benchmark for evaluating the faithfulness of multimodal large language models (MLLMs) in incorporating visual cues during reasoning for visual question-answering tasks. The authors draw data from the M3CoT and MegaBench datasets, extract key visual cues from images, modify them so that the correct answer changes, and generate new images based on the updated cues using GPT-Image-1. This setup enables testing whether models genuinely rely on visual information rather than recalling memorized solutions. The authors evaluate a broad range of open and proprietary models on VFaith-Bench, providing insights into the degree of visual reasoning faithfulness across different MLLMs.

**Strengths:**

1. The paper tackles an important and timely problem: assessing whether MLLMs can faithfully incorporate visual cues into their reasoning process is crucial for advancing multimodal reasoning research. This topic is of clear interest to the ICLR community.
2. VFaith-Bench provides a well-targeted diagnostic benchmark for systematically studying visual faithfulness in MLLMs.
3. The benchmark includes 755 samples spanning six task categories, and the semi-automated image-editing pipeline could be potentially applied to generate additional data in the future.
4. The evaluation covers a diverse set of open and proprietary models, with analysis conducted from multiple perspectives, including performance on original versus edited images and auxiliary metrics capturing memorization and perceptual grounding.
5. The paper is relatively clearly written and easy to follow.

**Weaknesses:**

1. The benchmark’s ability to reveal memorization effects is inherently limited. It can only expose potential memorization if the evaluated models were trained on data overlapping with M3CoT and MegaBench, which together contain around 20k instances. The authors do not verify whether these datasets were part of the training corpora of the evaluated open-source models. If they were not, the benchmark’s capacity to probe faithfulness issues related to memorization would be significantly reduced.
2. The authors acknowledge limitations of the GPT-Image-1–based editing pipeline in Section 3.2.1 and attempt to mitigate them through manual review. However, the analysis of “Reason 3” errors in Section 4.3 reveals that roughly 10% of the dataset still contains low-quality images that were missed in human filtering. The authors then remove these samples and recompute results (the “Refined” columns in Tables 2-3). This finding undermines confidence in the overall data quality and suggests that many reported results may be affected by this issue. Moreover, this crucial limitation appears only late in Section 4.4, disrupting the narrative flow and potentially misleading readers about dataset reliability.
3. The design of the Perception task raises questions. It is framed as a multi-choice task rather than as a set of binary classification subtasks, each dedicated to a specific visual cue. To me, the latter formulation would arguably be more well-defined and interpretable, allowing models to focus on isolated factors.
4. The Repeat Ratio metric is an interesting idea but conceptually underdeveloped and insufficiently explained. As currently defined, it measures—among cases where the model was initially correct—the proportion of incorrect answers (after image editing) that repeat the original correct answer. The metric only considers examples where the model was correct before editing, which biases comparisons across models with different base accuracies. It also ignores cases where models consistently repeat wrong answers, which could equally indicate reliance on memorization.
5. In the Evaluation Pipeline, the authors state that direct output matching was applied only for certain multiple-choice questions, while in other cases, evaluation was conducted using Claude-3.5-Sonnet. However, the paper does not specify which dataset categories use which evaluation protocol. Moreover, crucial details about the LLM-as-a-judge setup are missing, making it difficult to assess the reliability of the reported results. For instance, were the model-based evaluations correlated with human judgments to confirm their validity? Without such evidence, the faithfulness of the results obtained through automated grading remains questionable.
6. The qualitative analysis section provides limited insight. The authors do not discuss performance trends across dataset categories. For example, models occasionally perform better on edited than on original images in certain tasks (e.g., CRE and TIF, as with Gemini-2.5-Pro on TIF), yet this outcome is not analyzed or explained. A category-level breakdown or error analysis would have improved interpretability.
7. The experimental methodology in Section 4.3 raises significant concerns. Model responses are graded by Gemini-2.5-Pro and subsequently verified using Claude-3.5-Sonnet. However, Gemini-2.5-Pro itself performs imperfectly on VFaith-Bench (see Table 2), which undermines its reliability as an evaluator of other models’ reasoning. Additionally, Claude-3.5-Sonnet is never evaluated on the benchmark, leaving its assessment accuracy entirely untested. To ensure validity, the authors should manually review a subset of responses, categorize error types, and quantify agreement between human and model-based judgments.

**Questions:**

1. M3CoT contains over 11K samples and MegaBench over 8K, yet VFaith-Bench includes fewer than 1K samples. Could the dataset be expanded further using your semi-automated pipeline?
2. How many human annotators were involved in the Image Quality Checking Pipeline? How were disagreements handled, and what was the inter-annotator agreement (e.g., Cohen’s κ)?
3. Did the authors verify whether M3CoT or MegaBench were part of the training data of the evaluated open-source models? If not, how do the authors ensure that observed memorization or faithfulness issues are not attributable simply to unseen data?
4. Which dataset categories were evaluated via direct output matching versus Claude-3.5-Sonnet judgment?
5. Why was the Perception task formulated as a multi-choice question instead of several binary classification tasks, each isolating a single visual cue?
6. Since Gemini-2.5-Pro and Claude-3.5-Sonnet were used for model grading, did the authors perform any human–LLM agreement analysis to confirm that this evaluation pipeline is reliable?

---

### Official Review · Reviewer_1ojn · 2025-10-28

**Soundness:** 2
**Presentation:** 2
**Contribution:** 2
**Rating:** 4
**Confidence:** 4

**Summary:**

Current MLLMs often achieve high performance on complex reasoning tasks, but it's unclear if this success comes from genuine visual reasoning or from exploiting brittle patterns and memorized data. Existing benchmarks lack the ability to quantitatively assess the visual fidelity of MLLMs' reasoning. This paper introduces VFaith-Bench, a new benchmark designed to evaluate how MLLMs reason based on visual information, rather than simply relying on memorized patterns from training data.

**Strengths:**

The paper introduces a unique cue-driven editing pipeline for generating multimodal benchmark data to induce hallucinations and probe reasoning chains. The evaluations revealed deficiencies in visual cue perception and adherence, as well as potential data leakage, providing insights for developing more reliable MLLMs.

Unlike prior multimodal reasoning benchmarks (e.g., HallusionBench), this work goes beyond assessing raw reasoning ability — it probes how much of the reasoning is truly grounded in visual cues rather than memorized patterns or prior context.

The experiments reveal consistent insights, e.g., performance degradation after cue modification, discrepancies between perception and reasoning, and evidence of data leakage or memorization. All of which are empirically meaningful and reproducible findings that advance understanding of multimodal reasoning mechanisms.

**Weaknesses:**

1. The paper does not specify how many visual cues are extracted for each image, which is crucial because the number and granularity of cues directly affect the subsequent image editing process.

2. After large models extract vision cues, there appears to be no human validation or quality control to verify whether these cues are accurate or relevant. Similarly, in Section 4.3, the identification of error reasons (Reason 1–3) is entirely model-driven, without human cross-checking. This lack of human validation could compromise the credibility of the experimental results.

3. Although the paper reports the overall dataset size (755 samples) and the distribution across task types, it does not provide detailed benchmark statistics.

4. Table 2 presents detailed metrics for each subset, while Table 3 (for smaller reasoning models) only reports overall accuracy and ∆, without per-subset breakdowns. It would be helpful to include fine-grained results to show whether smaller models exhibit consistent performance degradation across all categories.

5. The paper is written in a rush, so there are many typos:

 in the abstract, "long-chain inference process contribute to its performance improvements" contribute --> contributes

line 204: we propose a MLLM reasoning output paradigm --> an MLLM

line 468: after refine--> after refinement

line 447: The proportion of errors answers --> error answers

**Questions:**

It is unclear why Section 4.4 (“Image Editing Quality Control and Refine”) is introduced as a separate stage. If low-quality edited images were detected, why were they not filtered out earlier during the human validation phase? The authors could have incorporated a rubric for image quality into the initial human validation process instead of conducting post-hoc filtering.

---

### Official Review · Reviewer_5C65 · 2025-11-01

**Soundness:** 4
**Presentation:** 3
**Contribution:** 3
**Rating:** 2
**Confidence:** 4

**Summary:**

This paper addresses the unclear link between MLLMs’ visual input and reasoning, and lack of metrics for visual faithfulness. It proposes: 1) A cue-driven auto-editing pipeline with GPT-Image-1 to modify key visual cues, building comparative QA pairs. 2) VFaith-Bench (755 entries, 6 subsets including a perception task) to evaluate MLLMs’ visual faithfulness in long reasoning. 3) A multi-model filtering mechanism plus manual checks to ensure image quality. Evaluations on closed-source and open-source models show all models drop in accuracy post-editing, with hallucinations and over-reliance on training patterns.

**Strengths:**

The major strengths are as follows:

S1. This paper is well written and is easy-to-follow.

S2. The studied task is interesting and important in the research field.

S3. The authors propose a good baseline to make data.

**Weaknesses:**

The weaknesses are clear from my point of view.

W1. From the methodology side. As a machine-generated benchmark, its quality and diversity are influenced by the machine. No new knowledge will be created.

W2. From the statistics side, some core details are also missed. For example, the distribution of the question, answers, comparisons with existing related benchmarks.

W3. The difficulty of the benchmark is limited. According to Tab. 2, the Gemini-2.5-pro can correctly answer 84.78% questions.

W4. Lack of a technical baseline to advance model performance.

**Questions:**

See weaknesses. In addition,

Q1. How will you distribute the benchmark to avoid data leakage?

Q2. How about using your data engine for model training?

---

### Official Review · Reviewer_dL7h · 2025-11-01

**Soundness:** 3
**Presentation:** 3
**Contribution:** 3
**Rating:** 4
**Confidence:** 4

**Summary:**

In this paper, the authors proposed VFaith-Bench, a benchmark to evaluate the visual faithfulness of Multimodal Large Language Models (MLLMs) when generating long reasoning process. Multiple experiments with several MLLMs show that the produced VFaith-Bench provide good challenges for both the open-source and proprietary models.

**Strengths:**

1. Both the motivation of evaluating MLLMs' faithfulness and the proposed VFaith-Bench make sense and are technically sound to me. Without a full ablation based benchmark like the one proposed in the draft, it is difficult to figure out the faithfulness of the responses from MLLMs.
2. The authors conducted extensive experiments on both open-source and proprietary models and show that they all suffer from the lack of visual faithfulness.
3. Writing is good and easy to follow.

**Weaknesses:**

1. My main concern is the heavy dependence on MLLMs themself in the curation of the benchmark. While it help automate the pipeline and make it more scalable, it is a bottleneck and capped by the capability of the models used.
2. The benchmark is limited to multiple-choice questions which is only a small portion of the whole spectrum of evaluations.

**Questions:**

Please refer to the paper weakness section for more details and provide more justification on the proposed benchmark.

---

### Note · Authors · 2025-11-12

I have read and agree with the venue's withdrawal policy on behalf of myself and my co-authors.